# Ser/Leu-swapped cell-free translation system constructed with natural/in vitro transcribed-hybrid tRNA set

Tomoshige Fujino [1], Ryogo Sonoda [1], Taito Higashinagata [1], Emi Mishiro-Sato [2], Keiko Kano [2] & Hiroshi Murakami [1,3] ✉

The Ser/Leu-swapped genetic code can act as a genetic firewall, mitigating biohazard risks arising from horizontal gene transfer in genetically modified organisms. Our prior work demonstrated the orthogonality of this swapped code to the standard genetic code using a cell-free translation system comprised of 21 in vitro transcribed tRNAs. In this study, to advance this system for protein engineering, we introduce a natural/in vitro transcribed-hybrid tRNA set. This set combines natural tRNAs from *Escherichia coli* (excluding Ser, Leu, and Tyr) and in vitro transcribed tRNAs, encompassing anticodon-swapped tRNA$^{Ser}_{GAG}$ and tRNA$^{Leu}_{GGA}$. This approach reduces the number of in vitro transcribed tRNAs required from 21 to only 4. In this optimized system, the production of a model protein, superfolder green fluorescent protein, increases to 3.5-fold. With this hybrid tRNA set, the Ser/Leu-swapped cell-free translation system will stand as a potent tool for protein production with reduced biohazard concerns in future biological endeavors.

Bacterial synthesis of proteins using a genetically modified organism (GMO) transformed with a gene from pathogenic organisms poses biohazard risks. A hazardous gene in the GMO could be transferred to other natural organisms through horizontal gene transfer[1–3]. Such inadvertent gene transfers have the potential to cause significant harm to human and animal health, as well as to the environment. Cell-free translation systems may provide a safer alternative for protein synthesis as they do not involve GMOs. Nonetheless, there remains a risk of gene leakage during the template DNA preparation process, which usually includes cloning genes from chemically synthesized DNA fragments or natural sources such as cDNA or genomic DNA into plasmids within a GMO.

To address this issue, we previously introduced the Ser/Leu-swapped genetic code (Supplementary Fig. 1)[4]. This genetic code allows for the synthesis of a protein without requiring the preparation of a hazardous gene. In this genetic code, Ser is assigned to Leu codons and vice versa, by utilizing chimeric tRNA$^{Ser}_{GAG}$ and tRNA$^{Leu}_{GGA}$, both of which have swapped anticodons. Genes encoded by the Ser/Leu-

swapped genetic code are not hazardous to natural organisms as they only produce Ser/Leu-swapped inactive proteins in nature.

However, the initial proof-of-concept version of the Ser/Leu-swapped cell-free translation system required the preparation of 21 tRNAs through in vitro transcription. This made the cell-free translation system impractical for protein synthesis. In addition, in vitro transcribed (IVT) tRNAs lack post-transcriptional modifications, while natural tRNAs extracted from *Escherichia coli* (*E. coli*) are reported to have, on average, 7.7 post-transcriptional modifications per molecule[5]. These modifications play diverse roles, including altering codon-amino acid correspondence[6,7], modulating decoding efficiencies of synonymous codons[8–10], promoting proper tRNA folding[11–16], and facilitating the efficient recognition of tRNAs by aminoacyl-tRNA synthetases such as IleRS, GluRS, and LysRS[17–19]. The production of proteins dropped to 20% when replacing the natural tRNA extract with the IVT tRNA set.

In this study, we create the hybrid-SL tRNA set. This set consists of tRNA$^{Ser}$, tRNA$^{Leu}$, and tRNA$^{Tyr}$-deficient natural tRNAs extracted from

[1]Department of Biomolecular Engineering, Graduate School of Engineering, Nagoya University, Nagoya, Japan. [2]Institute of Transformative Bio-Molecules (WPI-ITbM), Nagoya University, Nagoya, Japan. [3]Institute of Nano-Life-Systems, Institutes of Innovation for Future Society, Nagoya University, Nagoya, Japan. ✉e-mail: murah@chembio.nagoya-u.ac.jp

*E. coli*, alongside IVT-tRNA$^{Phe}$, tRNA$^{Tyr}$, chimeric tRNA$^{Ser}_{GAG}$, and chimeric tRNA$^{Leu}_{GGA}$. Utilizing this hybrid tRNA set, we establish a Ser/Leu-swapped cell-free translation system (Fig. 1) that achieves protein production efficiency comparable to a translation system using natural tRNAs.

## Results and discussion

### Construction of the natural/ IVT-hybrid tRNA set

*E. coli* has 37 tRNAs that decode 61 codons into 20 proteinogenic amino acids. Among these, tRNA$^{Ser}$, tRNA$^{Leu}$, and tRNA$^{Tyr}$ are longer (85−93 bases) than the other tRNAs (74−77 bases) (Fig. 2a). This characteristic allows us to prepare a tRNA$^{Ser}$ and tRNA$^{Leu}$-deficient natural tRNA set without having to isolate each individual tRNA.

Initially, we analyzed the *E. coli* natural tRNA extract using denaturing PAGE containing urea (Fig. 2a, lane 1). The mobility of band I was consistent with that of IVT tRNA$^{Ser}$, tRNA$^{Leu}$, and tRNA$^{Tyr}$ (Fig. 2a, lanes 3−5). Thus, we inferred that this band represents the natural tRNA$^{Ser}$, tRNA$^{Leu}$, and tRNA$^{Tyr}$. The Band II tRNAs were excised from the gel and purified (Fig. 2a, lane 2), and this tRNA mixture was defined as the ΔSLY natural tRNA set.

Subsequently, we prepared IVT tRNA$^{Tyr}$, chimeric tRNA$^{Ser}_{GAG}$, and chimeric tRNA$^{Leu}_{GGA}$ using T7 RNA polymerase. We then combined these with the ΔSLY natural tRNA set to assemble the hybrid tRNA set for the Ser/Leu-swapped genetic code, which we termed the hybrid-SL tRNA set (Fig. 2b). Similarly, we assembled the hybrid-Std tRNA set for the standard genetic code using IVT tRNA$^{Ser}$ and tRNA$^{Leu}$ instead of chimeric tRNAs. We also assembled a tRNA set with 21 IVT tRNAs for the Ser/Leu-swapped genetic code, termed the IVT21-SL tRNA set. Using one of these three tRNA sets, we prepared reconstituted cell-free translation systems for further study (Supplementary Fig. 2).

### Synthesis of peptides in a cell-free translation system with the hybrid-Std tRNA set

To verify the accurate decoding of codons in the cell-free translation system using the hybrid-Std tRNA set, we first assessed the synthesis of P1 peptides (Met-Tyr-Tyr-Tyr-Xaa-Asp-Asp-Arg-Asp, where Xaa represents 17 types of amino acids; Fig. 3a). Matrix-assisted laser desorption/ionization-time of flight mass spectrometry (MALDI-TOF MS) analysis demonstrated that all peptides were synthesized in the cell-free translation system (Fig. 3b top and Supplementary Figs. 3a and 4; calculated and observed masses are listed in Supplementary Table 1 and 2). Tricine SDS-PAGE analysis followed by autoradiography ([$^{14}$C]-Asp labeling) showed comparable amounts of peptides synthesized across all samples (Fig. 3c and Supplementary Fig. 3b). These results suggest that the natural tRNA species corresponding to 17 natural amino acids in the ΔSLY natural tRNA set accurately decode the Xaa codon. Tyr insertion was unexpectedly observed for peptides with Glu, Phe, Trp, and Lys. This anomaly might be due to translational slippage on repeated Tyr codons preceding the Xaa codon. Various translational slippages, such as −1, −4, and +2 frameshifts at the NNA-AAC-AAG sequence in an *E. coli* cell-free translation system[20], as well as +2 frameshift at the GUG-UG sequence and +6 ribosome hopping at the GUG-UGA-GUU sequence in *E. coli*[21], have been reported. In addition, the depletion of an aminoacyl-tRNA that corresponds to an A-site codon can increase the −1 frameshifting in an *E. coli* cell-free translation system[22]. This suggests that the unexpected insertion of Tyr observed in the mass spectra might be due to the low concentration of aminoacyl-tRNAs of Glu, Phe, Trp, and Lys (*vide infra*).

### Accuracy of Ser/Leu-swapping in a cell-free translation system with the hybrid-SL tRNA set

The presence of natural tRNA$^{Ser}$ and tRNA$^{Leu}$ in the hybrid-SL tRNA set could compromise accurate translation in the Ser/Leu-swapped cell-free translation system, leading to the synthesis of proteins that contain both Ser and Leu at the same positions. Although PAGE analysis

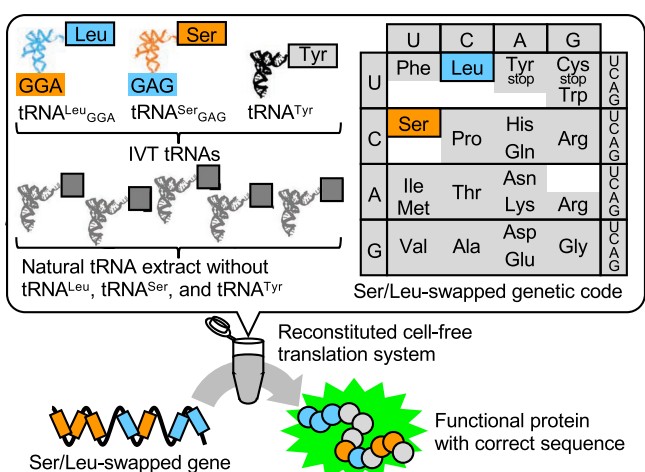

**Fig. 1 | Schematic representation of the concept of the Ser/Leu-swapped genetic code with the hybrid-SL tRNA set.** tRNA$^{Ser}$, tRNA$^{Leu}$, and tRNA$^{Tyr}$ are excluded from the *E. coli* natural tRNA extract. Chimeric tRNA$^{Leu}_{GGA}$, chimeric tRNA$^{Ser}_{GAG}$, and tRNA$^{Tyr}$ synthesized by in vitro transcription are added to the tRNA$^{Ser}$, tRNA$^{Leu}$, and tRNA$^{Tyr}$-excluded extract to prepare a hybrid-SL tRNA set. Using this hybrid tRNA set, a cell-free translation system with the Ser/Leu-swapped genetic code is constructed, where Ser is assigned to Leu codons and Leu is assigned to Ser codons. A functional protein is synthesized from a Ser/Leu-swapped gene of interest. Abbreviations: IVT in vitro transcribed, *E. coli Escherichia coli*.

showed no detectable amount of natural tRNA$^{Ser}$ and tRNA$^{Leu}$ in the ΔSLY natural tRNA set (Fig. 2a, lane 2), we verified the accuracy of Ser/Leu-swapping by translating P1 peptides (Met-Tyr-Tyr-Tyr-Xaa-Asp-Asp-Arg-Asp, where Xaa represents amino acid at UCU or CUU codons; Fig. 3a).

We synthesized model peptides with Ser or Leu at the Xaa position using a cell-free translation system with the hybrid-SL tRNA set. MALDI-TOF MS analysis confirmed the correct assignment of Ser and Leu to their respective swapped codons in the Ser/Leu-swapped genetic code without observable cross-contamination (Fig. 3b middle and Supplementary Table 1). Tricine SDS-PAGE analysis supported this result, as translated products showed a single intense band for each mRNA (Fig. 3c).

To determine the necessity of excluding natural tRNA$^{Ser}$ and tRNA$^{Leu}$ from the natural tRNA extract, we also tested a mix of the natural tRNA extract with chimeric tRNA$^{Ser}_{GAG}$ and tRNA$^{Leu}_{GGA}$. MALDI-TO FMS (Fig. 3b bottom and Supplementary Table 1) and tricine SDS-PAGE (Fig. 3c) detected mixtures of peptides with Ser and Leu, confirming the importance of excluding the natural tRNA$^{Ser}$ and tRNA$^{Leu}$ for the accurate Ser/Leu-swapping in the cell-free translation system.

### Optimization of the hybrid tRNA set for protein translation

Next, we aimed to synthesize a model protein in the cell-free translation system using the hybrid-Std tRNA set. We utilized a standard sfGFP gene encoding superfolder GFP (sfGFP) according to the standard genetic code and measured sfGFP production by spectrofluorometry (Supplementary Fig. 5, *gfp-Std-22C*). Unexpectedly, the fluorescence intensity of sfGFP synthesized with the hybrid-Std tRNA set was below 5% of that synthesized with the natural tRNA extract (Supplementary Fig. 6a).

We hypothesized that the concentration of specific natural tRNA(s) might have decreased during the PAGE purification process. To identify these tRNA(s), we supplemented the cell-free system with IVT tRNAs. We divided 18 IVT tRNAs into four groups (Groups A−D) and evaluated the sfGFP fluorescence intensity. Notably, the addition of Group A tRNAs (including tRNA$^{Phe}$, tRNA$^{Lys}$, tRNA$^{Trp}$, tRNA$^{Cys}$, and tRNA$^{Ala}$) substantially increased sfGFP production (Supplementary Fig. 6a). Further testing of individual tRNAs from Group A revealed that

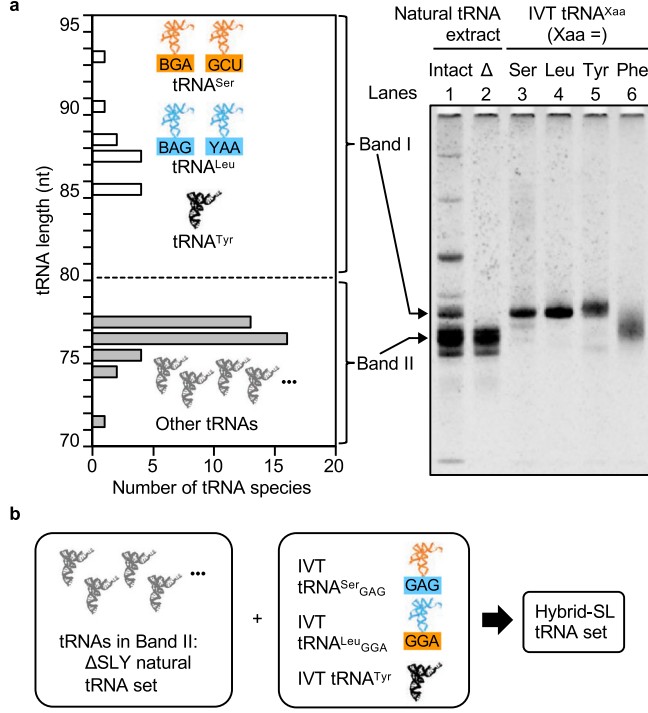

**Fig. 2 | Preparation of the hybrid-SL tRNA set. a** Exclusion of tRNA^Ser and tRNA^Leu from the *E. coli* natural tRNA extract. The length of *E. coli* natural tRNAs is shown in the graph on the left. White bars represent tRNAs with 85 or more nucleotides, specifically tRNA^Ser, tRNA^Leu, and tRNA^Tyr. In contrast, gray bars represent other tRNAs with 77 or less nucleotides. Through denaturing PAGE, these two tRNA groups can be distinguished into two separate bands: band I and band II. tRNAs found in band II were subsequently extracted from the gel and are referred to as the ΔSLY natural tRNA set. The right panel shows the denaturing PAGE analysis of the natural tRNA extract, ΔSLY natural tRNA set, and IVT tRNAs (*n* = 1). Lanes 1 and 2 represent the tRNA extract before and after gel purification, labeled as intact and ΔSLY natural tRNA set, respectively. Lanes 3–6 represent IVT tRNA^Ser, tRNA^Leu, tRNA^Tyr, and tRNA^Phe. The gel image was cropped only for this figure. **b** Construction of the hybrid-SL tRNA set. The ΔSLY natural tRNA set and IVT tRNAs, including chimeric tRNA^Ser_GAG, chimeric tRNA^Leu_GGA, and tRNA^Tyr were combined to prepare the hybrid-SL tRNA set. Abbreviations: IVT in vitro transcribed, *E. coli Escherichia coli*. Source data are provided as a Source Data file.

only tRNA^Phe significantly boosted sfGFP production, to about 78% of the level achieved with the natural tRNA extract (Supplementary Fig. 6b). Denaturing PAGE analysis showed a smeared band for IVT tRNA^Phe (Fig. 2a, lane 6), indicating partial loss of tRNA^Phe during the PAGE purification. Consequently, we included IVT tRNA^Phe in the hybrid tRNA sets for subsequent experiments.

### Ser/Leu-swapped synthesis of sfGFP using the hybrid-SL tRNA set

To study the synthesis of the model protein sfGFP in the Ser/Leu-swapped cell-free translation system, we prepared a Ser/Leu-swapped sfGFP gene encoded according to the Ser/Leu-swapped genetic code (Supplementary Fig. 5, *gfp-SL-22C*). Native PAGE analysis showed that the amount of active sfGFP produced with the hybrid-SL tRNA set was comparable to that with the natural tRNA extract (Fig. 4a, lane 4 vs. 1). Furthermore, the amount of active sfGFP was significantly improved to 3.5-fold compared to that with the IVT21-SL tRNA set (Fig. 4a, lane 4 vs. 5). This result indicates that the cell-free translation system with the hybrid-SL tRNA set is useful for in vitro protein production.

We also studied the orthogonality of the Ser/Leu-swapped genetic code and the standard genetic code by translating the protein using mismatched tRNA set/gene combinations: i.e., the Ser/Leu-swapped sfGFP gene with the natural tRNA extract and the standard sfGFP gene

with the hybrid-SL tRNA set. No active sfGFP was detected in either cell-free translation system (Fig. 4a, lanes 2 and 3). This lack of activity is ascribed to 30 mutations, comprising 10 Ser to Leu substitutions and 20 Leu to Ser substitutions (Supplementary Fig. 5), present in the translation products from mismatched tRNA set/gene combinations. Considering sfGFP is composed of 239 amino acids, the 30 residue Ser/Leu swaps constitute 13% of the protein. Autoradiography after native PAGE showed bands of different mobilities than active sfGFP, presumed to be misfolded or aggregated translation products (Fig. 4a, lanes 7 and 8). This hypothesis was supported by SDS-PAGE analysis, which showed a single band for each sample (Supplementary Fig. 7).

### Comparison of hybrid-SL and IVT21-SL tRNA sets for translating genes encoded by 22 or 47 sense codons

Using the IVT21-SL tRNA set, only 35 of the 61 sense codons (Supplementary Fig. 2) are available to encode a protein, as the tRNA set consists of 21 IVT tRNAs. In contrast, 53 of the 61 sense codons are available to encode a protein when using the hybrid-SL tRNA set, as the tRNAs for 16 standard amino acids (excluding Ser, Leu, Tyr, and Phe) were isolated from the natural tRNA extract (Supplementary Fig. 2).

For this study, we used a Ser/Leu-swapped sfGFP gene containing 22 sense codons (Supplementary Fig. 5, *gfp-SL-22C*). To verify codon availability, we prepared another Ser/Leu-swapped sfGFP gene containing 47 sense codons (excluding rare codons) (Supplementary Fig. 5, *gfp-SL-47C*). The synthesis of inactive proteins in a cell-free translation system using the natural tRNA extract from this gene demonstrated the orthogonality of the Ser/Leu-swapped genetic code with 53 sense codons against the standard genetic code (Supplementary Fig. 8).

In the cell-free translation system with the hybrid-SL tRNA set, a similar amount of active sfGFP was synthesized from both genes (Fig. 4b, lanes 1, 2, 5, and 6), suggesting that at least 47 sense codons are available to encode a protein. In contrast, while sfGFP was synthesized from the Ser/Leu-swapped sfGFP gene containing 22 sense codons (Fig. 4b, lanes 4 and 8), none was produced from the gene containing 47 sense codons (Fig. 4b, lanes 3 and 7) in the cell-free translation system using the IVT21-SL tRNA set. This is attributed to the IVT21-SL tRNA set's lack of tRNAs to decode 14 codons, representing 26% of the coding sequence, in the *gfp-SL-47C* gene.

In addition, we developed a translation system using a hybrid-SL tRNA set from an *E. coli* strain with tRNA genes for rare codons. This hybrid-SL tRNA set enhanced sfGFP production from a gene carrying 15 rare codons (Supplementary Fig. 9, *pa-gfp-SL-RC15* and Supplementary Fig. 10). Recent research indicates that the codon choice impacts amino acid incorporation accuracy[23] and protein folding by modulating translation rates[24,25] for some proteins. Therefore, the cell-free translation system with the hybrid-SL tRNA set is potentially invaluable for gene design.

### Ser/Leu-swapped synthesis of streptavidin and β-galactosidase using the hybrid-SL tRNA set

To demonstrate the synthesis of another model protein in the Ser/Leu-swapped cell-free translation system, we prepared a Ser/Leu-swapped streptavidin gene encoded according to the Ser/Leu-swapped genetic code (Supplementary Fig. 11, *stv-SL*). Before native PAGE analysis, fluorescent-labeled biotin was added to detect active streptavidin. The result showed that the amount of active streptavidin in the cell-free translation system with the hybrid-SL tRNA set was comparable to that with the natural tRNA extract (Supplementary Fig. 12, lane 4 vs. 1). When attempting to translate proteins using mismatched tRNA set/gene combinations, no active streptavidin was detected (Supplementary Fig. 12, lanes 2 and 3), consistent with the results from the sfGFP experiments. The cause of this result is attributed to 22 Ser/Leu-swapped codons in the gene constituted with 160 codons. A similar outcome was noted for a Ser/Leu-swapped β-galactosidase gene, with

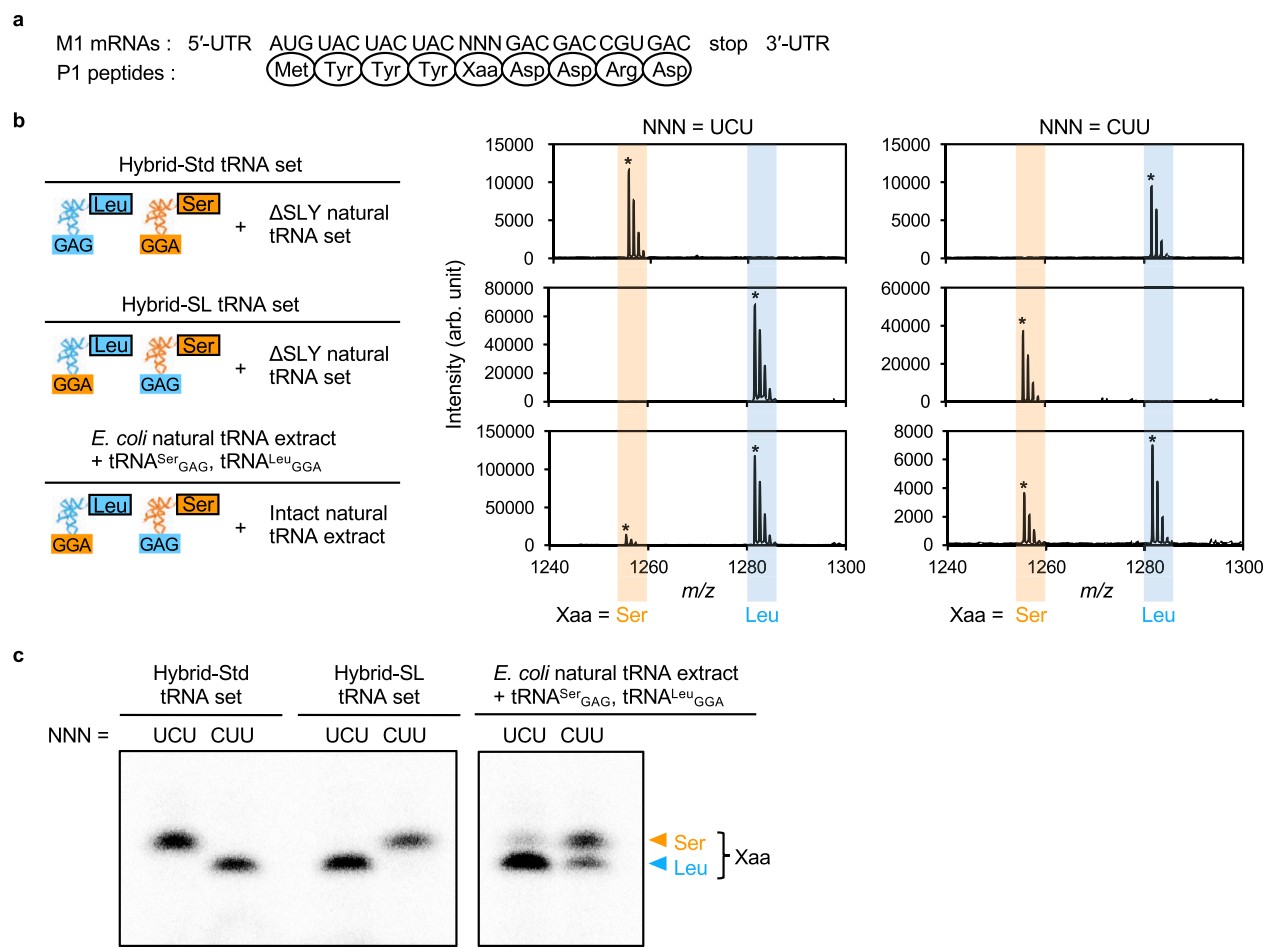

**Fig. 3 | Translation of the model peptides containing Ser and Leu codons by using a cell-free translation system with a hybrid tRNA set. a** The sequence of the model peptides and their coding mRNAs. Xaa and NNN represent the amino acids Ser and Leu, and the codons UCU and CUU, respectively. **b** MALDI-TOF MS analysis of the model peptides. Peaks corresponding to the full-length model peptides containing either Ser or Leu are labeled with asterisks in the MS spectra. The full MS spectra are shown in Supplementary Fig. 4. The calculated and observed mass of the model peptides are provided in Supplementary Table 1. **c** Tricine SDS-PAGE analysis of the model peptides. The peptides were analyzed using tricine SDS-PAGE and detected by autoradiography ($n = 1$). Samples were run on the same gel, and the image was cropped only for the purpose of this figure. Abbreviations: UTR, untranslated region; Std, standard; SL, Ser/Leu-swapped. Source data are provided as a Source Data file.

a total of 156 Ser/Leu-swapped codons within a 1040-codon gene (Supplementary Fig. 13, *pa-gal-SL*; Supplementary Fig. 14), though the protein production levels were too low for SDS-PAGE detection. These experiments conclusively demonstrated the orthogonality of the Ser/Leu-swapped genetic code in the synthesis of other proteins.

## LC-MS analysis of peptides and proteins produced in the cell-free translation systems

To further examine the products synthesized in cell-free translation systems, we used liquid chromatography-mass spectrometry (LC-MS) analysis. We prepared M2 and M3 mRNAs, each containing CU(U/C) or UC(U/C) codons, to investigate the effects of amino acid mis-incorporation (Fig. 5) and frame-shifting on these mRNAs (Supplementary Fig. 15). In experiments with these mRNAs, peptides P2 and P3, which contain Ser at the Xaa position, were predominantly produced from mRNAs with CU(U/C) codons when using the hybrid-SL tRNA set (Fig. 5, middle row; Supplementary Table 3 and 4; Supplementary Figs. 16–25 for MS/MS data), aligning with findings presented in Fig. 3. A minor presence of Leu-containing P2 and P3 peptides was noted, likely due to residual tRNA$^{Leu}$ in the ΔSLY natural tRNA set or the small amount of tRNA$^{Leu}$ remaining in the purified ribosome and EF-Tu solutions used to prepare the cell-free translation system[26]. This by-product was also detected in samples without the chimeric tRNA$^{Ser}_{GAG}$

(Fig. 5, bottom row; Supplementary Table 3 and 4; Supplementary Figs. 16–25 for MS/MS data). Notably, misreading led to P2 and P3 peptides containing Phe or Thr, and Pro or Thr, respectively, suggesting misreading at specific codon/anticodon pairs (Fig. 5, bottom row; Supplementary Table 3 and 4; Supplementary Figs. 16–25 for MS/MS data; Supplementary Fig. 26a). The relative abundance of these by-products varied between M2 and M3 mRNAs, indicating that misreading efficiency might be influenced by the mRNA context around the CU(U/C) codons. Similarly, for mRNAs with UC(U/C) codons, peptides P2 and P3 predominantly contained Leu, with Ser-containing peptides appearing as minor by-products, again likely due to residual tRNA$^{Ser}$ in the ΔSLY set or the purified ribosome and EF-Tu solutions. This pattern was also observed without chimeric tRNA$^{Leu}_{GGA}$. Misreading also led to the incorporation of Phe or Thr in the P2 peptide, and Pro or Thr in the P3 peptide (Fig. 5, bottom row; Supplementary Table 3 and 4; Supplementary Figs. 16–25 for MS/MS data), attributed to mispairing at specific codon/anticodon interfaces (Supplementary Fig. 26b). The frequency of misreading products was also different between M2 and M3 mRNAs, which again suggested that the misreading frequency could depend on the mRNA context surrounding the CUU/C codons. The misincorporated amino acids in these codons align with previous research by Kato et al.[27]. Our results further emphasize that the frequency of codon misreading could be

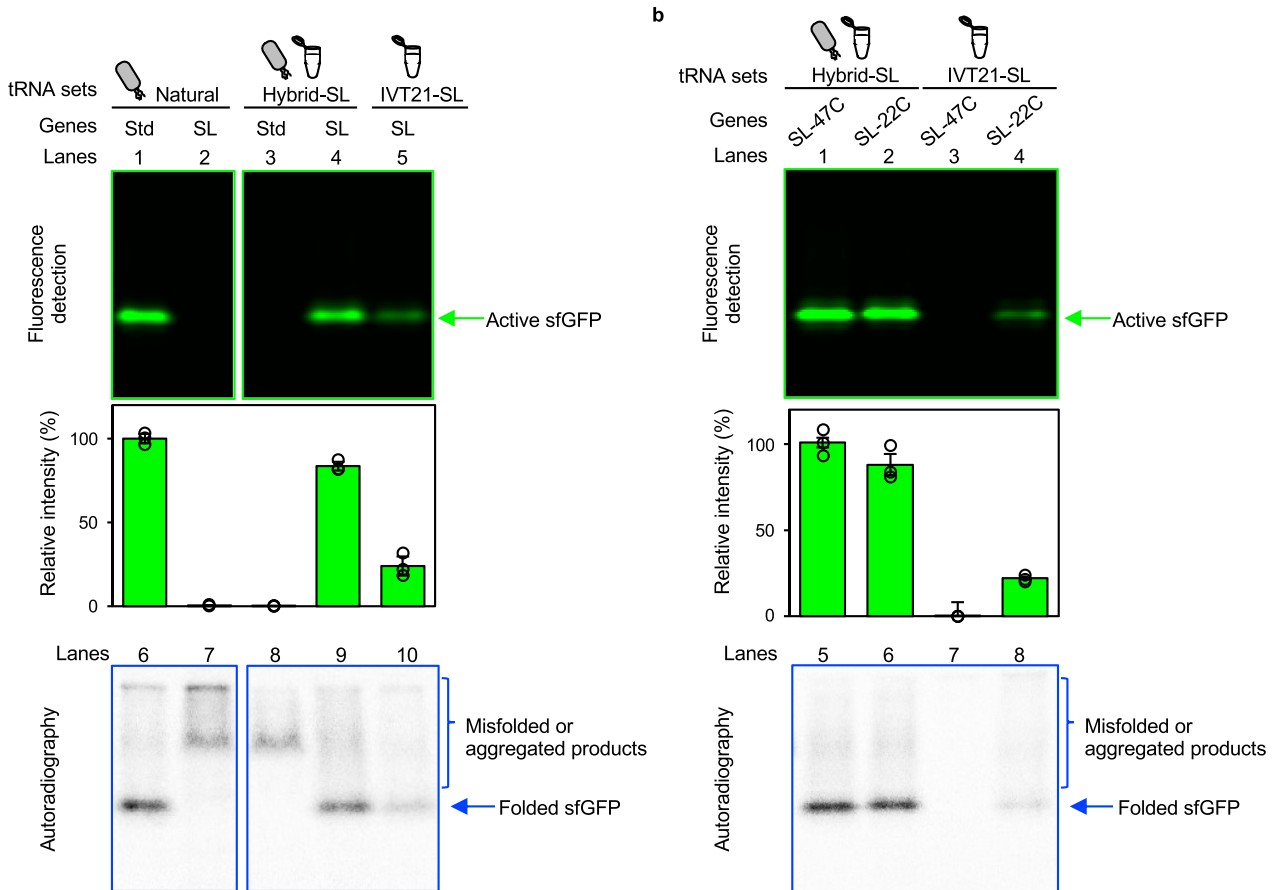

**Fig. 4 | Analysis of active sfGFP expression. a** Comparison of three tRNA sets (natural tRNA extract, hybrid-SL tRNA set, and IVT21-SL tRNA set) for the translation of the sfGFP. Cell-free translation systems were constructed using one of the tRNA sets, and one of the sfGFP genes coded with the standard genetic code (*gfp-Std-22C* gene) or the Ser/Leu-swapped genetic code (*gfp-SL-22C* gene) were translated. Samples were run on the same gel, and the image was cropped only for the purpose of this figure. **b** Comparison of the hybrid-SL tRNA set and the IVT21-SL tRNA set in translating a Ser/Leu-swapped sfGFP genes containing 22 (*gfp-SL-22C* genes) or 47 sense codons (*gfp-SL-47C* genes). The upper panel shows the result of native PAGE followed by fluorescence imaging of sfGFP. Gel images were cropped only for

this figure. Gel images are representative of *n* = 3 biologically independent experiments. Bars represent mean ± SD, and open circles represent individual data points for *n* = 3 biologically independent experiments. The relative band intensities were normalized against the intensity of the band observed in the combination of the natural tRNA extract and *gfp-Std-22C* gene (Fig. 4a, lane 1). For quantitative comparisons between samples on different gels, the same standard sample was applied to both gels. The bottom panel displays the autoradiography result of the same gel. Abbreviations: IVT in vitro transcribed, Std standard, SL, Ser/Leu-swapped, sfGFP superfolder green fluorescent protein. Source data are provided as a Source Data file.

influenced by the mRNA context, as previously observed with UAG misreading[28]. Furthermore, Thr misincorporation persisted in translations of M3 mRNAs with UC(U/C) and CU(U/C) codons, potentially due to the suboptimal performance of seryl-tRNA synthetase and the heightened misreading frequency influenced by the mRNA context around these codons.

We also identified peptides generated from −2, −1, +1, or +3 frameshifts (Fig. 6 and Supplementary Figs. 27–30 for MS/MS data). The peptide with a −2 frameshift could be produced through the mis-interaction of tRNA$^{Leu}_{GAG}$ with the UUC codon followed by a −2 frameshift that rearranges it to a canonical codon-anticodon pair, CUU-GAG (Supplementary Figs. 31, f1). The three peptides with a −1 frameshift could be produced through misinteraction of tRNA$^{Thr}_{(V/G)GU}$ with the CU(U/C) codon followed by a −1 frameshift that rearranges it to a codon-anticodon pair, UCU-(V/G)GU (Supplementary Figs. 31, f2–4). The other three peptides with a −1 frameshift could be produced through the interaction of tRNA$^{Pro}_{GGG}$ with the CCC codon followed by a −1 frameshift that also results in a canonical codon-anticodon pair, CCC-GGG (Supplementary Figs. 31, f5–7). The two peptides with a +1 frameshift could be produced through the interaction of tRNA$^{Pro}_{VGG}$ with the CCC codon followed by a +1 frameshift that also results in a

canonical codon-anticodon pair, CCG-VGG (Supplementary Figs. 31, f8–9). Three peptides with +1 or +3 frameshifts were identified in the translation system with the hybrid-SL tRNA set without tRNA$^{Leu}_{GGA}$, presumably to avoid the unassigned UCU codon (Supplementary Figs. 31, f10–12).

Given the reported loss of accurate decoding by chimeric tRNA$^{Leu}_{(U/C)GA}$ (misreading near-cognate UCU codon)[29] and tRNA$^{Ala}_{CGA}$ (misreading near-cognate UCA codon)[30], we analyzed the misreading products from M2 mRNAs containing near-cognate codons (Supplementary Fig. 32). After adding chimeric tRNA$^{Ser}_{GAG}$ and tRNA$^{Leu}_{GGA}$ to a translation system with the natural tRNA extract, we examined the products via LC-MS (Supplementary Figs. 33–35). Among them, we identified three peptides with Leu or Ser substitution, whose relative abundance against the original peptides increased upon the addition of the chimeric tRNAs. The relative abundance of the P2-4 peptide with Phe to Ser substitution, which could result from the misreading of the UUC near-cognate codon by the chimeric tRNA$^{Ser}_{GAG}$ through a U/G mismatch at the first base, was increased from 0.004 to 0.12. In addition, the relative abundance of the P2-1 peptide with a Pro-to-Ser substitution increased from 0.012 to 0.037, and a Pro-to-Leu substitution increased from 0.001 to 0.011. These peptides could be

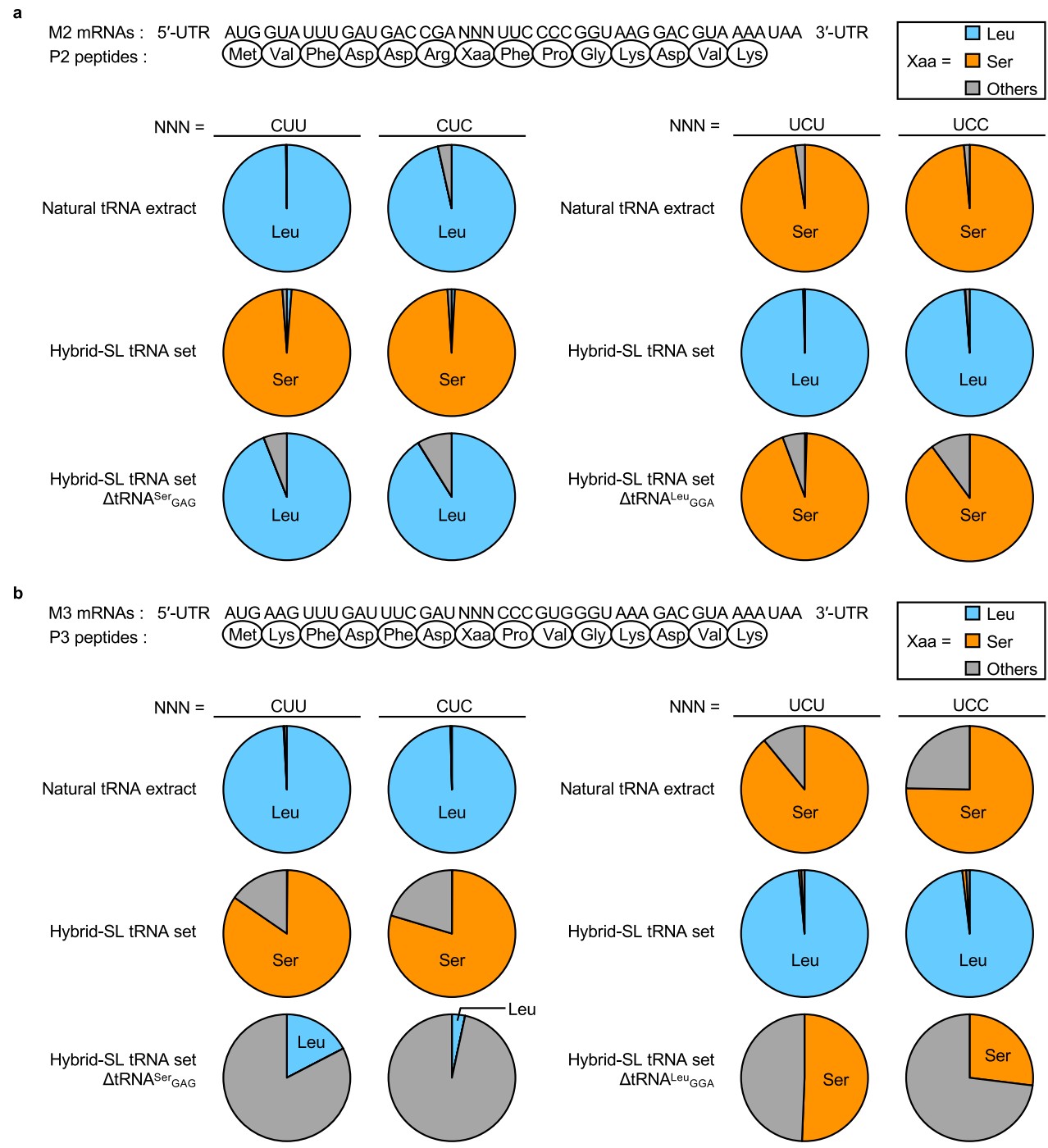

**Fig. 5 | LC-MS analysis of the products synthesized in a cell-free translation system containing either the natural tRNA extract, the hybrid-SL tRNA set, or the hybrid-SL tRNA set minus one chimeric tRNA. a** The P2 peptide sequences and their encoding M2 mRNAs are displayed, with NNN denoting the CUU/C and UCU/C codons. The relative abundance of amino acids (Xaa) identified via LC-MS is presented in the pi charts. The relative abundance was calculated from the abundance of each peptide divided by the total abundance of observed peptides. The relative abundance of amino acids is presented in Supplementary Table 3. **b** The P3 peptide sequences and their encoding M3 mRNAs. The relative abundance of amino acids (Xaa) is presented in the pi charts. The relative abundance of amino acids is presented in Supplementary Table 4. Source data are provided as a Source Data file.

produced by the misreading of the CCC near-cognate codon by the chimeric tRNA$^{Ser}_{GAG}$ through a C/A mismatch at the second base or by the chimeric tRNA$^{Leu}_{GGA}$ through a C/A mismatch at the first base.

Finally, LC-MS analysis was conducted on proteins (Supplementary Fig. 36, *pa-gfp-Std* and *pa-gfp-SL*; Supplementary Fig. 37, *pa-stv-Std* and *pa-stv-SL*; Supplementary Fig. 13, *pa-gal-Std* and *pa-gal-SL*) synthesized from Ser/Leu-swapped genes to assess amino acid incorporation at the swapped codons. The analysis of model proteins purified using anti-PA tag antibody beads post-trypsin digestion yielded MS/MS data of five fragments encoded by DNA sequences with one Ser/Leu-swapped codon (Supplementary Figs. 38–40). The main product in the translation mixture with the hybrid-SL tRNA set contained

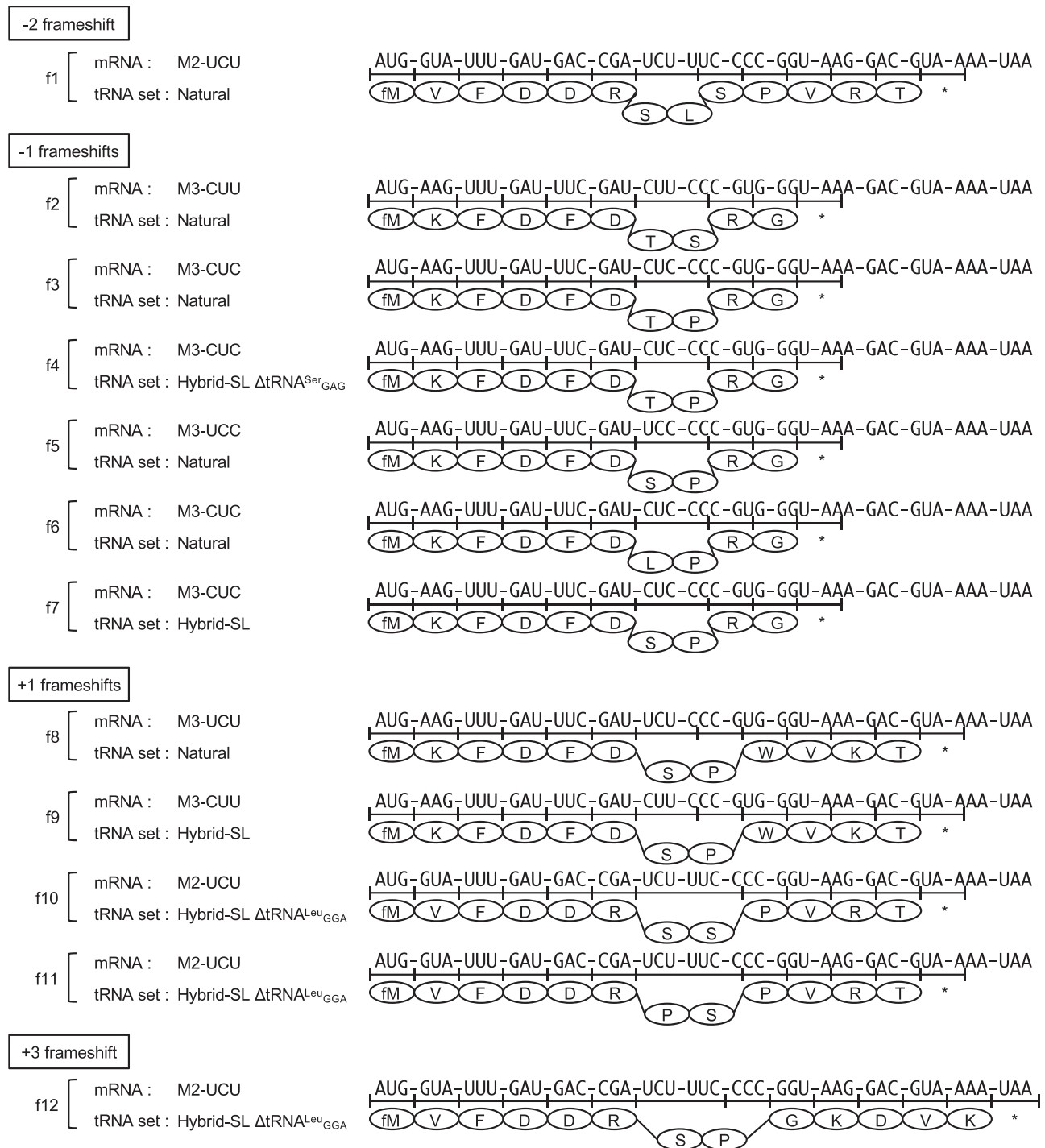

**Fig. 6 | Frameshift products observed in the LC-MS analysis of the peptides synthesized in a cell-free translation system containing either the natural tRNA extract, the hybrid-SL tRNA set, or the hybrid-SL tRNA set minus one chimeric tRNA.** Observed peptides produced by frameshifts in the M2 and M3 mRNAs are displayed. The possible mechanisms of frameshifts are presented in Supplementary Fig. 31. Abbreviations: UTR, untranslated region; SL - Ser/Leu-swapped.

Leu at the UCU codons (GFP_135, streptavidin_171, β-galactosidase_41, and 424) and Ser at the CUU codon (GFP_102). Leu incorporation at the GFP_102 CUU codon was also observed, likely due to residual tRNA^Leu in the ΔSLY natural tRNA set.

In summary, we successfully developed a cell-free translation system with the Ser/Leu-swapped genetic code, employing the hybrid-SL tRNA set. This hybrid tRNA set enhanced sfGFP production, achieving a 3.5-fold increase compared to the previously reported IVT21-SL tRNA set. This not only improves translation efficiency but

also reduces the associated effort needed to prepare the tRNA set for constructing the Ser/Leu-swapped genetic code. Importantly, the majority of the natural tRNAs are retained in the hybrid-SL tRNA set, making 53 out of 61 sense codons available for designing Ser/Leu-swapped genes.

Notably, the chimeric tRNAs have been observed to elevate misreading rates of the UUC and CCC near-cognate codons, likely due to a U/G mismatch at the first or the second base, resulting in the incorporation of Ser and Leu instead of Phe or Pro. To accurately decode the

remaining eight codons, seven distinct IVT chimeric tRNAs must be employed within the cell-free system. This requirement might affect translation fidelity, as these IVT chimeric tRNAs potentially exhibit reduced accuracy. Investigating the precision of these chimeric tRNAs is essential to fully develop the complete Ser/Leu-swapped genetic code. For instance, tRNA engineering such as optimizing the anticodon bases[31] or implementing post-transcriptional modifications[5–19] might be necessary to diminish misreading. Alternatively, this error could be mitigated by finely tuning the concentration of each aminoacyl-tRNA within the system. Exploring the overall translation fidelity and the balance of aminoacyl-tRNA concentrations would provide valuable insights into how cells uphold their translational machinery.

Our findings also indicate that the production of peptides with frameshifts and those containing incorrect amino acids at the UCU/C and CUU/C positions is influenced by the surrounding mRNA sequences. This highlights the importance of careful gene design to avoid translation errors. Future research should explore how the context of mRNA sequences affects frameshifts and the incorporation of incorrect amino acids. In addition, our translation system exhibited a relatively high incidence of peptides or proteins with misincorporated amino acids. Given that our system is reconstituted, it may be missing key components that promote accurate translation. Identifying these missing elements could significantly improve the utility of the in vitro translation system, facilitating more effective protein production.

Our study also demonstrated that proteins translated from Ser/Leu-swapped model protein genes were inactive when translated in the cell-free translation system with the standard genetic code. This finding underscores the reduced risk of gene leakage to natural organisms when employing Ser/Leu-swapped genes. Conversely, our findings point to the increasing necessity for cyber biosecurity. To counter the potential chemical synthesis of DNA encoding dangerous genes, a guideline was established by the International Gene Synthesis Consortium[32]. This protocol involves rejecting the synthesis of any gene identified as hazardous through sequence screening within a database of pathogenic genomes. Yet, genes designed with a codon-swapped genetic code could potentially evade such screenings. Therefore, the adaptation of sequence screening techniques to identify codon-swapped patterns of hazardous genes will become increasingly crucial for cyber-biosecurity.

The concept of an orthogonal genetic code, especially one involving amino acid-swapping, has attracted significant attention[33,34] since our report in 2020[4]. Recently, Chin's group reported on the prevention of a mobile genetic element transfer by reassigning the Ser UC(A/G) codons to Ala, His, or Pro using chimeric tRNA$^{Ala}_{(U/C)GA}$, tRNA$^{His}_{(U/C)GA}$, or tRNA$^{Pro}_{(U/C)GA}$ in a codon compressed *E. coli*[30,35]. Church's group also reported that the prevention of phage replication by reassigning the Ser UC(A/G) codons to Leu using chimeric tRNA$^{Leu}_{(U/C)GA}$[29]. In addition to in vivo approaches from other groups, our cell-free method offers promising avenues to mitigate future biohazard risks.

## Methods
### Materials
*E. coli* Rosetta™ 2 (DE3) strain (cat no. TRNAMRE-RO) and the natural tRNA extract derived from *E. coli* MRE600 strain (cat no. 71397) were purchased from Merck. Anti-PA tag antibody beads (cat no. 012-25841) were purchased from FUJIFILM Wako Pure Chemical Corporation. Synthetic DNAs containing *gfp-Std-22C*, *gfp-SL-22C*, *pa-gfp-SL-RC5*, *pa-gfp-SL-RC9*, *pa-gfp-SL-RC15*, *stv-Std*, *stv-SL*, *pa-gal-Std*, *pa-gal-SL*, *pa-gfp-Std*, *pa-gfp-SL*, *pa-stv-Std*, and *pa-stv-SL* genes were purchased from Integrated DNA Technologies and Genscript. A plasmid containing the *gfp-SL-47C* gene was purchased from Eurofins Genomics. Oligonucleotides were purchased from FASMAC. M1, M2, and M3 mRNAs encoding model peptides (Figs. 3a, 5, and Supplementary Data 1) and

mRNAs encoding model proteins (Supplementary Figs. 5, 9, 11, 13, 36, and 37, and Supplementary Data 2) were prepared with the procedure described in Supplementary Method 1 (see Supplementary Data 3 for primer sequences). 21 IVT tRNAs (Supplementary Table 5) were also prepared with the procedure described in Supplementary Method 2 (see Supplementary Data 4 and 5 for primer sequences).

### Preparation of ΔSLY natural tRNAs
A denaturing PAGE gel was prepared with acrylamide/bisacrylamide (37.5:1, 10%), 6 M urea, and TBE (44.5 mM tris·borate pH 8.0, 2 mM EDTA) in a large gel cassette [height = 15 cm, width = 15 cm, gel thickness = 2 mm, 1 well (depth = 2 cm, width = 5 cm)]. A loading sample containing the natural tRNA extract (200 μL; 50 mg/mL tRNA extract, 4 M urea, 12.5 mM EDTA pH 8.0) was applied to the gel. Electrophoresis was carried out at 300 V for 100 min at room temperature. Band II (Fig. 2a) was cut out, and then crushed using a mortar and pestle. The crushed gel was resuspended in 15 mL of 0.3 M NaCl, and the solution was shaken overnight at room temperature to elute the tRNAs. The gel debris was removed by centrifugation, and twice the volume of ethanol was added to the supernatant. The solution was incubated at −20 °C for 30 min and then centrifuged (4 °C, 15,000 × *g*, 15 min). After removing the supernatant, the precipitate was dissolved in 1 mL of ultra-pure water. After filtering through a 45 μm syringe filter (Millipore, cat no. SLCRX13NK) and treating with phenol/chloroform, tRNAs were pelleted by ethanol precipitation. The resultant pellet was air-dried and redissolved in 100 μL of ultra-pure water. This tRNA mixture was termed the ΔSLY natural tRNA set. Both the ΔSLY natural tRNA set and the 10 mg/mL natural tRNA extract were analyzed using denaturing PAGE (6 M urea, 10% acrylamide). The gel was stained with ethidium bromide, imaged using the Gel Doc EZ system (Bio-Rad), and analyzed with Image Lab software (Bio-Rad). The ΔSLY natural tRNA set stock solution was prepared by adding ultra-pure water to achieve the same concentration of band II tRNAs as in the 10 mg/mL natural tRNA extract. The concentration of the ΔSLY natural tRNA set stock solution was determined by OD at 260 nm. To construct the ΔSLY natural tRNA set for genes containing rare codons, total tRNA was extracted from *E. coli* strain Rosetta™ 2 (DE3) cells following Supplementary Method 3[36].

### Preparation of tRNA stock solutions
The compositions of the IVT tRNA mix are listed in Supplementary Table 6. The tRNA stock solutions including the ΔSLY natural tRNA set and the natural tRNA extract were heated at 95 °C for 5 min and then cooled to 25 °C to refold tRNAs before the translation reaction.

### Preparation of *E. coli* reconstituted cell-free translation systems
*E. coli* reconstituted cell-free translation systems were prepared in our previous work[37]. The composition of the translation system is listed in Supplementary Table 7. In general, the hybrid tRNA translation mixture (5 μL) containing the 0.6 mg/mL ΔSLY natural tRNA set and 10 μM of each IVT tRNA was prepared by adding 0.75 μL of the 4 mg/mL ΔSLY natural tRNA set stock solution and 0.5 μL of the IVT tRNA mix (100 μM each). The translation mixture also contained 0.5 mM of each of the 19 amino acids (except Asp) and 2 μM of mRNA. The translation reaction was performed at 37 °C for 3 hours. For peptide synthesis, 50 μM [$^{14}$C]-Asp (Moravek, cat no. MC139) for autoradiography, 50 μM non-radioactive Asp for MALDI-TOF MS, or 500 μM non-radioactive Asp for LC-MS was used. For protein synthesis, a mixture of 50 μM [$^{14}$C]-Asp and 150 μM non-radioactive Asp for autoradiography, 500 μM non-radioactive Asp for other experiments was used.

The translation mixture (5 μL) containing 1.5 mg/mL of the natural tRNA extract was prepared by adding 0.75 μL of the 10 mg/mL natural tRNA extract. The IVT21-SL translation mixture (5 μL) containing 10 μM of each IVT tRNA (21 kinds of IVT tRNAs) was prepared by adding 2 μL of the IVT tRNA mix (25 μM each) following Supplementary Method 2[4].

## Analysis of translated products

The synthesized model peptides labeled with [$^{14}$C]-Asp in the translation mixture were analyzed using 15% polyacrylamide tricine SDS-PAGE followed by autoradiography (Pharos Fx imager, Bio-Rad). Alternatively, reactions with Asp instead of [$^{14}$C]-Asp were performed. Reaction mixtures (2 μL) were applied to SPE C-TIP KT200 columns (Nikkyo Technos), which had been equivalated by 15 μL of elution solution (80% acetonitrile, 0.5% acetic acid). The columns were washed with 15 μL of wash solution (4% acetonitrile, 0.5% acetic acid) twice. Peptides were eluted by 2 μL of matrix solution (saturated α-cyano-4-hydroxycinnamic acid, 80% acetonitrile, 0.5% acetic acid) and applied onto the MALDI sample plate. After dry-up, the samples were analyzed by MALDI-TOF MS (autoflex maX, Bruker).

The synthesized model proteins labeled with [$^{14}$C]-Asp in the translation mixture were analyzed by 15% polyacrylamide native PAGE and SDS-PAGE followed by autoradiography (Pharos Fx imager, Bio-Rad). For sfGFP analysis, the gel was also analyzed by fluorescence detection (λex = 473 nm, LPB filter; Typhoon FLA 9000, GE Healthcare).

For streptavidin analysis, 2.5 μL of 2 μM Atto 488-biotin (Sigma-Aldrich, cat no. 30574) was added to 2.5 μL of the reaction mixture to label active streptavidin with fluorophore. The excess Atto 488-biotin was removed using streptavidin M280 magnetic beads (Thermo Fisher Scientific, cat no. 60210) before subjecting the samples to native PAGE. The gel after native PAGE was analyzed by fluorescence detection (ChemiDoc MP, Bio-RAD).

For β-galactosidase analysis, 2.5 μL of 100 μM Fluorescein di-β-D-galactopyranoside (Abcam, cat no. ab273643) was added to 2.5 μL of the translation mixture. After incubation at 37 °C for 30 min, the solution was diluted 40 times with 25 mM Hepes-K pH7.5, and 150 mM NaCl and analyzed using fluorescence spectrometry (FP-8500, JASCO).

For optimization of the hybrid Std-tRNA set, the translation mixture (5 μL) containing synthesized sfGFP was diluted five times with 50 mM Tris-HCl pH8.0 and analyzed using fluorescence spectrometry (FP-8500, JASCO).

LC-MS/MS analysis of the translated products is described in the Supplementary Method 4 (see Supplementary Data 6 for the list of the FASTA and data files).

## Reporting summary

Further information on research design is available in the Nature Portfolio Reporting Summary linked to this article.

## Data availability

LC-MS data generated in this study have been deposited to ProteomeXchange with the identifier PXD050604. Source data are provided with this paper.

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

## Acknowledgements
This study was financially supported by a Grant-in-Aid for Scientific Research(S) (grant number 23H05456 to H.M.), and a Grant-in-Aid for Scientific Research(C) (grant number 21K05270 to T.F.) from the Japan Society for the Promotion of Science and a donation from H. Murakami.

## Author contributions
Experiments were designed by T.F. and H.M. Samples were prepared by T.F., R.S., T.H. and H.M. Data except LC-MS analysis were acquired by T.F. and R.S. LC-MS data were acquired by E.M.S. and K.K. Figure design and manuscript writing were performed by T.F. and H.M. The project was supervised by H.M.

## Competing interests
The authors declare no competing interests.
