## [Peer Review File · Nature Communications]

Ser/Leu-swapped cell-free translation system constructed with natural/in vitro transcribed-hybrid tRNA setReviewers' Comments:

Reviewer #1:

Remarks to the Author:

Fujino et. al describe an engineered system which swaps Ser and Leu codons to create an environment that helps mitigate future biohazard risks from working with genetically modified materials. The authors have done a thorough job to characterize and compare their system to standard cell-free translation and other variations of the hybrid system. While the data presented regarding production of sfGFP is very convincing, in the world of genetic engineering the question always is how robust these systems are for other protein sequences. The impact of this work would be increased by using this system to produce another protein with the hybrid-SL tRNA set.

Minor suggestions:

Line 445: Missing space "hybrid-SLtRNA"

Figure 3 vs 4: Why is one 50 μ M Asp and the other 500 μ M?

Lines 151-154: This sentence is confusing and hard to grasp what the authors are trying to say.

Reviewer #2:

Remarks to the Author:

This work by Fujino et al. describes an elegant, simplified way of generating in vitro amino acid swapped genetic codes to prevent horizontal gene transfer. This is an important research direction because artificial genetic codes, especially the swapped code described in the text, can efficiently block incoming and outgoing gene transfer. These in vitro systems can provide a rapid test for follow-up in vivo studies, can test more extreme genetic code refactoring than what's possible with current recoded cells, and allow a more detailed characterization of translational effects that otherwise could remain hidden in living cells. The authors' work is a significant step towards these aims. However, I found it questionable whether the results presented in the current version of the manuscript are sufficient to demonstrate a significant advance compared to their 2020 paper (10.1021/acssynbio.0c00196) and fully support the authors' goal of orthogonality and "reduced biohazard concerns in future biological endeavors". Especially considering recent results demonstrating that in vitro results do not directly translate into in vivo functional amino acid swapped genetic codes (PMIDs 36264827 and 36922599).

My main concern is related to the authors' definition of orthogonality. Orthogonality and translation accuracy are both essential qualities of gene-transfer-free artificial genetic codes, but a recent in vivo study highlighted the reduced fidelity of a similar Leu tRNA-based swapped code (36264827) and the proteomics-based characterization in 36922599 showed low-level mistranslations at Ser codons not targeted by the anticodon-swapped Leu tRNAs. While these subtle effects can remain hidden when the orthogonality of codon reassignments is only investigated using less sensitive methods (e.g., the MALDI-TOF MS on single codon positions or GFP fluorescence), slight mistranslation events quickly add up and compromise the purity and function of the final protein product. In in vivo systems, this mistranslation can lead to fitness degradation or the lack of viability.

My suggestion for the authors is to revise the manuscript's analytical sections and provide additional evidence for orthogonality and fidelity using more sensitive assays:

- First, the authors should demonstrate that their PAGE-based tRNA isolation workflow removes even traces of the natural serine and leucine tRNAs. Or if traces of natural tRNAs remain, these should not compromise the translation fidelity of the final hybrid translation mixture. The authors' PAGE- and imaging-based assay is not sensitive enough. This experiment can be done more quantitatively by performing an in vitro translation reaction using the Δ SLY natural tRNA set (lacking tRNA^{Ser}, tRNA^{Leu}, and tRNA^{Tyr}) and then translating model peptides containing mRNAs with a single instance of every codon corresponding to the deleted tRNA^{Ser}, tRNA^{Leu}, and tRNA^{Tyr} tRNAs, and then performing HPLC-MS and mapping the resulted data to all potential peptides that can be translated from the given

mRNA. The analysis should also include the detection of truncated and extended, e.g., *ssrA*-tagged products, which are expected if the tRNA removal functions well. The ratio of natural-sequence-like and erroneous peptides should be presented in the manuscript. A similar assay is described in *eLife* 2018;7:e34878 and 10.1016/j.bbagen.2017.01.025.

- Next, with the reconstituted natural/IVT-hybrid tRNA set for the Ser/Leu-swapped genetic code, the authors should measure translation accuracy simultaneously at multiple instances of the targeted Ser and Leu codons in multiple genetic contexts (i.e., mRNAs with multiple different reassigned codons and using multiple distinct mRNA sequences) using proteomics approaches, i.e., tryptic digest followed by HPLC-MS, instead of only MALDI or functional assays. This is necessary because, as the authors also show in line 87, codons preceding and surrounding the decoded codon influence translation fidelity and codon decoding. The analysis of a single mRNA with a single codon position and MALDI MS-based readout cannot detect minute amounts of mistranslated products and characterize the effect of translation in different codon contexts.

In addition to the targeted Ser and Leu codons, measuring mistranslation efficiency at codons not directly recognized by the IVT anticodon-swapped tRNAs (i.e., codons with one mismatch in their 1st, 2nd, and especially in the 3rd nucleotide position compared to the targeted codons) is also necessary considering the mistranslation effects showed by others in the papers referenced above.

- Finally, the same proteomics-based characterization (tryptic digest followed by HPLC-MS instead of MALDI or only functional assays) should also be performed on the *in vitro*-produced sfGFP. The authors should analyze the amino acid composition at the reassigned codon positions using proteomics-based peptide identification and rule out the presence of near-cognate suppression and contaminating amino acids at the reassigned codons.

- It would also be beneficial to see a wide range of other proteins synthesized using the authors' swapped code because the synthesis of sfGFP was already demonstrated in their 2020 paper. It was challenging to evaluate the utility of the authors' *in vitro* system when the product is only a single small protein with only a few reassigned codons. What is the upper limit in the number of distinct proteins that can be simultaneously synthesized using this system?

- A minor but important point for applicability is the authors' result in line 154. They note that mistranslation using the SL code results in various protein fragments. However, if the swapped code functions as expected, only a single product with equal length and mobility with the wild type sfGFP is expected. I encourage the authors to characterize the composition of these mistranslation byproducts using proteomics and describe why these truncated products appear.

Additional comments:

In the final section after line 158 the authors describe that 53 codons are available for protein production, however, they only demonstrate the functionality of a subset of these codons channels. The omitted rare codons can be easily supplemented using IVT tRNAs and a larger codon pool makes gene-design easier. Rare codons are useful for optimizing ribosomal binding sites, reducing repeat-mediated instability and can expedite DNA synthesis. I encourage the authors to extend this final section and show the benefits of their new *in vitro* code compared to their 2020 study.

The Conclusion section, in its current form, lacks an important analysis of the study's limitations and potential implications for biosafety. The possibility of horizontal gene transfer from *in vitro* expression systems to living organisms is limited, but the authors' swapped code can potentially compromise existing sequence screening systems, including the ones used by DNA synthesis companies or the International Gene Synthesis Consortium (PMID: 33247280) and potentially allow the production of toxins and other harmful proteins. I encourage the authors to describe these in their conclusion.

Lastly, as the tRNA expression constructs, gene sequences, and MS data are central to the manuscript's main conclusions, I hope the authors will list these sequences in their revised version and deposit all Mass Spectrometry data in public repositories.

Reviewer #3:

Remarks to the Author:

This study entitled "Ser/Leu-swapped cell-free translation system constructed with natural/in vitro transcribed-hybrid tRNA set" developed a new technique to swap ser and leu codons in a cell-free translation system. The authors' new idea is to separate natural tRNA mixtures based on their sizes, obtaining tRNA mixture that lacks ser, leu, and tyr-tRNAs (deltaSLY tRNA mixture). The deficient tRNA mixture was utilized for making a codon-swapped cell-free gene expression by combining chimeric leu-tRNA and ser-tRNA. I feel that the results presented here are worth reporting for researchers concerning biosafety. The experiments are sounds and the description is precise. However, I have some concerns about the importance of this study and the lack of experimental detail as described below.

Major points

1. I'm not convinced by the authors' claim that this study contributes to reducing biohazard concern. If the authors swapped Ser/Leu for in vivo system, I agree that this system contributes to biosafety. However, the authors' system is a cell-free system, which never grows and propagates in nature. Also, researchers usually use a cell-free system in a laboratory, which means that there is almost no chance for the used genes to be transferred to other organisms. What kind of risks do the authors think are associated with a cell-free system and why do they think Ser/Leu swapping is necessary? I think an explanation is needed in the Introduction and Discussion section.
2. Related to the above point, a large shortcoming of this manuscript is insufficient introduction (especially in the first paragraph) and no discussion. It was hard for me to understand the importance of this study with the current manuscript.
3. I could not find the explanation for the cell-free system used here. Is it a cell extract or a reconstituted system from purified components? Is it an E. coli system? I could not find any explanation in the manuscript. The authors wrote that it was prepared using a procedure similar to that described in our previous work (ref 29)(lane 266). In the reference, the authors use a reconstituted system, but the preparation procedure was not described. In the later part (lane 278), the author wrote, "The translation mixture for the IVT-SL tRNA set was prepared according to the procedure described in our previous report (ref 4)". In that paper, the authors seemed to use an E. coli cell extract. Which is right?
4. Lane 143-146. The authors compared GFP expression between the IVT-SL tRNA and the hybrid-SL tRNA. Are the total tRNA concentrations the same for both tRNA sets? I could not find the description of the IVT tRNA concentrations.

Minor point

1. Lane 65. Please describe how the author harvested the tRNAs in band II in the main text.

Responses to the referees' comments

We would like to thank the referees for their useful comments. We have addressed them all in the revised manuscript. Please find below a point-by-point answer to each comment.

Reviewer #1

Comment: While the data presented regarding production of sfGFP is very convincing, in the world of genetic engineering the question always is how robust these systems are for other protein sequences. The impact of this work would be increased by using this system to produce another protein with the hybrid-SL tRNA set.

Answer: We additionally applied the SL-swapped translation system to two other proteins (streptavidin and β -galactosidase), and the results were added as Supplementary Figs. 12 and 13 and described in lines 194-210.

Comment: Figure 3 vs 4: Why is one 50 μ M Asp and the other 500 μ M?

Answer: In our standard cell-free translation system, each amino acid was present at a final concentration of 500 μ M. However, when incorporating [14 C]-Asp (with a stock concentration of 495 μ M), we reduced the total Asp concentration to 50 μ M. This adjustment was necessary to strengthen the radiolabeled peptide signal as the addition of non-radioactive Asp could outcompete [14 C]-Asp.

Comment: Lines 151-154: This sentence is confusing and hard to grasp what the authors are trying to say.

Answer: We have rephrased the text for clarity in lines 156-159.

Comment: Line 445: Missing space "hybrid-SLtRNA"

Answer: Thank you for pointing out the error; we have rectified it.

Reviewer #2

Comment: This work by Fujino et al. describes an elegant, simplified way of generating *in vitro* amino acid swapped genetic codes to prevent horizontal gene transfer. This is an important research direction because artificial genetic codes, especially the swapped code described in the text, can efficiently block incoming and outgoing gene transfer. These *in vitro* systems can provide a rapid test for follow-up *in vivo* studies, can test more extreme genetic code refactoring than what's possible with current recoded cells, and allow a more detailed characterization of translational effects that otherwise could remain hidden in living cells. I found it questionable whether the results presented in the current version of the manuscript are sufficient to demonstrate a significant advance compared to their 2020 paper (10.1021/acssynbio.0c00196) and fully support the authors' goal of orthogonality and "reduced biohazard

concerns in future biological endeavors". Especially considering recent results demonstrating that *in vitro* results do not directly translate into *in vivo* functional amino acid swapped genetic codes (PMIDs 36264827 and 36922599). My main concern is related to the authors' definition of orthogonality. Orthogonality and translation accuracy are both essential qualities of gene-transfer-free artificial genetic codes, but a recent *in vivo* study highlighted the reduced fidelity of a similar Leu tRNA-based swapped code (36264827) and the proteomics-based characterization in 36922599 showed low-level mistranslations at Ser codons not targeted by the anticodon-swapped Leu tRNAs. While these subtle effects can remain hidden when the orthogonality of codon reassignments is only investigated using less sensitive methods (e.g., the MALDI-TOF MS on single codon positions or GFP fluorescence), slight mistranslation events quickly add up and compromise the purity and function of the final protein product. In *in vivo* systems, this mistranslation can lead to fitness degradation or the lack of viability. My suggestion for the authors is to revise the manuscript's analytical sections and provide additional evidence for orthogonality and fidelity using more sensitive assays.

Answer: We think that the translation system utilizing the hybrid-SL tRNA set surpasses the previously published IVT21-tRNA set from 2020 as an *in vivo* research model, due to its composition predominantly of natural tRNAs. We agree with Reviewer #2 on the significance of both *in vitro* and *in vivo* studies for examining the characteristics of amino acid-swapped genetic codes. Accordingly, we have conducted additional experiments, including LC-MS analysis, and incorporated these into the revised manuscript (please see below responses), which should provide valuable information for subsequent research.

Comment: First, the authors should demonstrate that their PAGE-based tRNA isolation workflow removes even traces of the natural serine and leucine tRNAs. Or if traces of natural tRNAs remain, these should not compromise the translation fidelity of the final hybrid translation mixture. The authors' PAGE- and imaging-based assay is not sensitive enough. This experiment can be done more quantitatively by performing an *in vitro* translation reaction using the Δ SLY natural tRNA set (lacking tRNA^{Ser}, tRNA^{Leu}, and tRNA^{Tyr}) and then translating model peptides containing mRNAs with a single instance of every codon corresponding to the deleted tRNA^{Ser}, tRNA^{Leu}, and tRNA^{Tyr} tRNAs, and then performing HPLC-MS and mapping the resulted data to all potential peptides that can be translated from the given mRNA. The analysis should also include the detection of truncated and extended, e.g., ssrA-tagged products, which are expected if the tRNA removal functions well. The ratio of natural-sequence-like and erroneous peptides should be presented in the manuscript. A similar assay is described in eLife 2018;7:e34878 and 10.1016/j.bbagen.2017.01.025. Next, with the reconstituted natural/IVT-hybrid tRNA set for the Ser/Leu-swapped genetic code, the authors should measure translation accuracy simultaneously at multiple instances of the targeted Ser and Leu codons in multiple genetic contexts (i.e., mRNAs with multiple different reassigned codons and using multiple distinct mRNA sequences) using proteomics approaches, i.e., tryptic digest followed by HPLC-MS, instead of only MALDI or functional assays. This is necessary because, as the authors also show in line 87, codons preceding and surrounding the decoded codon influence translation fidelity and codon decoding. The analysis of a single mRNA with a single codon

position and MALDI MS-based readout cannot detect minute amounts of mistranslated products and characterize the effect of translation in different codon contexts.

Answer: Pursuant to Reviewer #2's recommendation, we conducted further experiments and included the results in Fig. 5 and Supplementary Figs. 15 and 16, which are described in lines 212-241. While a small quantity of natural tRNA^{Leu} and tRNA^{Ser} persisted in the Δ SLY natural tRNA set, possibly decoding the corresponding Ser/Leu-swapped codons, this unwanted decoding was significantly reduced with the hybrid-SL tRNA set. This indicates the negligible role of the remaining natural tRNA^{Leu} and tRNA^{Ser}. Moreover, we noticed amino acid misincorporation at the S/L swapped codons. Interestingly, this misincorporation sometimes persisted with the hybrid-SL tRNA set, as well as with the natural tRNA extract, depending on the context of the surrounding mRNA sequence. We could also obtain data about frameshifts, which was added as Fig. 5c and Supplementary Fig. 18. The result was described in lines 242-256. We did not find any other products, including ssrA-tagged products, and the by-products resulted in a loss of translational fidelity as observed in eLife 2018; 7:e34878.

Comment: In addition to the targeted Ser and Leu codons, measuring mistranslation efficiency at codons not directly recognized by the IVT anticodon-swapped tRNAs (i.e., codons with one mismatch in their 1st, 2nd, and especially in the 3rd nucleotide position compared to the targeted codons) is also necessary considering the mistranslation effects showed by others in the papers referenced above.

Answer: Following the suggestion from Reviewer #2, we undertook additional experiments and integrated the findings as Supplementary Fig. 21, with details provided in lines 257-270. The chimeric tRNA^{Ser}_{GAG} increased the misreading of the UUC and CCC near-cognate codons and did not result in a detectable rise in misreading products from mRNAs containing other near-cognate codons. Similarly, the chimeric tRNA^{Leu}_{GGA} increased the misreading of the CCC near-cognate codon.

Comment: the same proteomics-based characterization (tryptic digest followed by HPLC-MS instead of MALDI or only functional assays) should also be performed on the *in vitro*-produced sfGFP. The authors should analyze the amino acid composition at the reassigned codon positions using proteomics-based peptide identification and rule out the presence of near-cognate suppression and contaminating amino acids at the reassigned codons.

Answer: Beyond sfGFP, we also conducted LC-MS analysis for streptavidin and β -galactosidase. These results have been added as Supplementary Fig. 22 and described in lines 271-278 in the main text.

Comment: It would also be beneficial to see a wide range of other proteins synthesized using the authors' swapped code because the synthesis of sfGFP was already demonstrated in their 2020 paper. It was challenging to evaluate the utility of the authors' *in vitro* system when the product is only a single small protein with only a few reassigned codons.

Answer: We adopted the SL-swapped translation system for the examination of two more proteins, streptavidin and β -galactosidase, and documented in Supplementary Figs. 12 and 13, as elaborated in

lines 194-210.

Comment: What is the upper limit in the number of distinct proteins that can be simultaneously synthesized using this system?

Answer: The constraints encountered originate from the reconstituted cell-free translation system. Previous research has demonstrated the co-expression of multiple proteins within such a system (e.g., ACS Synth Biol. 2017;6:1327–1336). Investigating the limits to the number of distinct proteins expressible is a prospective research direction.

Comment: A minor but important point for applicability is the authors' result in line 154. They note that mistranslation using the SL code results in various protein fragments. However, if the swapped code functions as expected, only a single product with equal length and mobility with the wild type sfGFP is expected. I encourage the authors to characterize the composition of these mistranslation byproducts using proteomics and describe why these truncated products appear.

Answer: As Reviewer #2 pointed out, native-PAGE analysis of Ser/Leu-swapped translation products showed the multiple bands. We think these bands are corresponded to the mis-folded or aggregated translation products because it was single band in SDS-PAGE. We added the SDS-PAGE as Supplementary Fig. 8.

Comment: In the final section after line 158 the authors describe that 53 codons are available for protein production, however, they only demonstrate the functionality of a subset of these codons channels. The omitted rare codons can be easily supplemented using IVT tRNAs and a larger codon pool makes gene-design easier. Rare codons are useful for optimizing ribosomal binding sites, reducing repeat-mediated instability and can expedite DNA synthesis. I encourage the authors to extend this final section and show the benefits of their new *in vitro* code compared to their 2020 study.

Answer: Instead of supplementing IVT tRNAs for the rare codons, we prepared the hybrid-SL tRNA set from Rosetta™ 2(DE3) cells, and demonstrated the translation of sfGFP from a gene containing 15 rare codons. We added the result in Supplementary Fig. 11, as elaborated in lines 187-192.

Comment: The Conclusion section, in its current form, lacks an important analysis of the study's limitations and potential implications for biosafety. The possibility of horizontal gene transfer from *in vitro* expression systems to living organisms is limited, but the authors' swapped code can potentially compromise existing sequence screening systems, including the ones used by DNA synthesis companies or the International Gene Synthesis Consortium (PMID: 33247280) and potentially allow the production of toxins and other harmful proteins. I encourage the authors to describe these in their conclusion.

Answer: We added the discussion about the study's limitations and potential implications for biosafety in the conclusion section (lines 288-320).

Comment: Lastly, as the tRNA expression constructs, gene sequences, and MS data are central to the manuscript's main conclusions, I hope the authors will list these sequences in their revised version and deposit all Mass Spectrometry data in public repositories.

Answer: We listed sequences of IVT-tRNAs and genes in Supplementary Tables 3 and 6, Supplementary Fig. 5, and the Source data file. We also deposit all Mass Spectrometry data to JPOST with the identifier JPST002984 (PXD050604). The URL and access key are as follows:

URL: <https://repository.jpostdb.org/preview/206410380465f24df83c41c>

Access key: 2176

Reviewer #3

Comment: I'm not convinced by the authors' claim that this study contributes to reducing biohazard concern. If the authors swapped Ser/Leu for *in vivo* system, I agree that this system contributes to biosafety. However, the authors' system is a cell-free system, which never grows and propagates in nature. Also, researchers usually use a cell-free system in a laboratory, which means that there is almost no chance for the used genes to be transferred to other organisms. What kind of risks do the authors think are associated with a cell-free system and why do they think Ser/Leu swapping is necessary? I think an explanation is needed in the Introduction and Discussion section. Related to the above point, a large shortcoming of this manuscript is insufficient introduction (especially in the first paragraph) and no discussion. It was hard for me to understand the importance of this study with the current manuscript.

Answer: We usually need to make GMO when we prepare a template DNA from chemically synthesized DNA fragments or natural sources of cDNAs or genome by cloning these gene in a plasmid. We added an explanation in the Introduction section in the revised manuscript (lines 32-36). We also added some sentences in the conclusion section (lines 288-320).

Comment: I could not find the explanation for the cell-free system used here. Is it a cell extract or a reconstituted system from purified components? Is it an *E. coli* system? I could not find any explanation in the manuscript. The authors wrote that it was prepared using a procedure similar to that described in our previous work (ref 29)(lane 266). In the reference, the authors use a reconstituted system, but the preparation procedure was not described. In the later part (lane 278), the author wrote, "The translation mixture for the IVT-SL tRNA set was prepared according to the procedure described in our previous report (ref 4)". In that paper, the authors seemed to use an *E. coli* cell extract. Which is right?

Answer: In this study, we used a *E. coli* reconstituted cell-free translation system. We added listed the composition of the translation system in Supplementary Table 4. We also rewrote the preparation of the *E. coli* reconstituted cell-free translation systems in the method section to explain more detail (lines 376-390).

Comment: Lane 143-146. The authors compared GFP expression between the IVT-SL tRNA and the hybrid-SL tRNA. Are the total tRNA concentrations the same for both tRNA sets? I could not find the

description of the IVT tRNA concentrations.

Answer: The total tRNA concentrations for the IVT-SL tRNA and the hybrid-SL tRNA is different. In the cell-free translation system carrying IVT-SL-tRNA set, the concentration of each *in vitro*-transcribed tRNA was adjusted to 10 μ M. Since the translation system included 21 tRNAs, the total final concentration of tRNA was 210 μ M, equivalent to approximately 5.3 mg/mL. On the other hand, the translation system carrying hybrid-SL-tRNA set included 10 μ M each of four *in vitro*-transcribed tRNAs and 0.6 mg/mL of Δ SLY natural tRNA set. Therefore, the total final concentration of tRNA was approximately 1.6 mg/mL. We renamed the the IVT-SL tRNA to the IVT21-SL tRNA to make the explanation clear. We also added the concentration of Δ SLY natural tRNA set in the method section (line 377).

Comment: Lane 65. Please describe how the author harvested the tRNAs in band II in the main text.

Answer: We rewrite the sentence as “The Band II tRNAs were excised from the gel and purified (Fig. 2a, lane 2), and this tRNA mixture was defined as the Δ SLY natural tRNA set.” in the revised manuscript (lines 70-71).

Reviewers' Comments:

Reviewer #1:

Remarks to the Author:

The authors have appropriately addressed the reviewer comments.

Reviewer #2:

Remarks to the Author:

Thank you for revising the manuscript. My comments were adequately addressed.

Reviewer #3:

Remarks to the Author:

My initial concerns have been resolved in the revised manuscript. Also, I think the newly introduced experiments significantly improve the quality of the paper. I have only one minor comment.

In lanes 222 and 231, the authors wrote that the minor peptide is likely due to residual tRNA^{Leu} or Ser in the delta-SLY set. Another possibility is residual natural tRNA in the purified EF-Tu and ribosome used in PURE system. A detectable level of tRNA remains in these purified components (see Miyachi et al. 2022 ACS Synthetic Biology, 2791–2799).

Responses to the referees' comments

We would like to thank the referees for reviewing our revised manuscript. According to Reviewer 3, we cited the article (Miyachi et al., 2022) and added the sentence describing it.

Reviewer #3

Comment: My initial concerns have been resolved in the revised manuscript. Also, I think the newly introduced experiments significantly improve the quality of the paper. I have only one minor comment.

In lanes 222 and 231, the authors wrote that the minor peptide is likely due to residual tRNA^{Leu} or Ser in the delta-SLY set. Another possibility is residual natural tRNA in the purified EF-Tu and ribosome used in PURE system. A detectable level of tRNA remains in these purified components (see Miyachi et al. 2022 ACS Synthetic Biology, 2791–2799).

Answer: Based on the reviewer's advice, we cited the article (Miyachi et al., 2022, ACS Synthetic Biology) and added a sentence describing the potential contamination of the Δ SLY tRNA set with residual natural tRNA in the purified EF-Tu and ribosome.